# Kidney Injury Caused by Preeclamptic Pregnancy Recovers Postpartum in a Transgenic Rat Model

**DOI:** 10.3390/ijms22073762

**Published:** 2021-04-05

**Authors:** Sarah M. Kedziora, Kristin Kräker, Lajos Markó, Julia Binder, Meryam Sugulle, Martin Gauster, Dominik N. Müller, Ralf Dechend, Nadine Haase, Florian Herse

**Affiliations:** 1Experimental and Clinical Research Center (ECRC), A Joint Cooperation of Charité—Universitätsmedizin Berlin and Max Delbruck Center for Molecular Medicine, 13125 Berlin, Germany; sarah.kedziora@mdc-berlin.de (S.M.K.); kristin.kraeker@charite.de (K.K.); lajos.marko@charite.de (L.M.); dominik.mueller@mdc-berlin.de (D.N.M.); ralf.dechend@charite.de (R.D.); nadine.haase@mdc-berlin.de (N.H.); 2Max Delbruck Center for Molecular Medicine (MDC) in the Helmholtz Association, 13125 Berlin, Germany; 3Charité-Universitätsmedizin Berlin, Corporate Member of Freie Universität Berlin and Humboldt-Universität zu Berlin, 10117 Berlin, Germany; 4Berlin Institute of Health at Charité–Universitätsmedizin Berlin, 10117 Berlin, Germany; 5DZHK (German Centre for Cardiovascular Research), Partner Site Berlin, 13353 Berlin, Germany; 6Department of Obstetrics and Gynecology, Medical University of Vienna, 1090 Vienna, Austria; julia.binder@meduniwien.ac.at; 7Institute for Clinical Medicine, Faculty of Medicine, University of Oslo, 0318 Oslo, Norway; uxsume@ous-hf.no; 8Division of Obstetrics and Gynaecology, Oslo University Hospital, 0450 Oslo, Norway; 9Division of Cell Biology, Histology and Embryology, Gottfried Schatz Research Centre, Medical University of Graz, 8036 Graz, Austria; martin.gauster@medunigraz.at; 10Helios Klinikum, 13125 Berlin, Germany

**Keywords:** preeclampsia, kidney injury, postpartum, transgenic rat model

## Abstract

Preeclampsia (PE) is characterized by the onset of hypertension (≥140/90 mmHg) and presence of proteinuria (>300 mg/L/24 h urine) or other maternal organ dysfunctions. During human PE, renal injuries have been observed. Some studies suggest that women with PE diagnosis have an increased risk to develop renal diseases later in life. However, in human studies PE as a single cause of this development cannot be investigated. Here, we aimed to investigate the effect of PE on postpartum renal damage in an established transgenic PE rat model. Female rats harboring the human-angiotensinogen gene develop a preeclamptic phenotype after mating with male rats harboring the human-renin gene, but are normotensive before and after pregnancy. During pregnancy PE rats developed mild tubular and glomerular changes assessed by histologic analysis, increased gene expression of renal damage markers such as kidney injury marker 1 and connective-tissue growth factor, and albuminuria compared to female wild-type rats (WT). However, four weeks postpartum, most PE-related renal pathologies were absent, including albuminuria and elevated biomarker expression. Only mild enlargement of the glomerular tuft could be detected. Overall, the glomerular and tubular function were affected during pregnancy in the transgenic PE rat. However, almost all these pathologies observed during PE recovered postpartum.

## 1. Introduction

Preeclampsia (PE) affects about 5–8% of all pregnancies worldwide, with a high maternal mortality rate due to complications such as eclampsia, HELLP-syndrome, or edema [1,2,3]. According to the International Society for the Study of Hypertension in Pregnancy, PE is characterized by the development of gestational hypertension (resting systolic blood pressure (BP) ≥ 140 and/or diastolic ≥ 90 mmHg) at or after 20 weeks of gestation in previously normotensive women [1]. Since proteinuria is present in only 75% of all cases [4], the American College of Obstetrics and Gynecology adapted guidelines for PE established in 2019 stated that “either proteinuria (>300 mg/L per 24-h urine) or other maternal organ dysfunction” should be present [1,5]. The underlying pathology of PE is not yet fully understood but a placental origin featuring immunological dysfunctions and vascular alterations plays an important role with a consequence of substantial long-term cardiovascular and metabolic risk to mothers and children [6,7,8,9]. Various studies report that PE is not only associated with acute kidney injury but can lead to substantial renal disease and failure postpartum as well as later in life [10,11,12]. The renal injury during human PE was first described by Spargo et al. as glomerular endotheliosis, meaning enlarged glomeruli with destruction of the capillary lumen [13], attributed to hypertrophy of endothelial or mesangial cells [14]. The presence of glomerular sclerosis, occlusion of capillaries due to thickening of the glomerular capillary wall, infiltration of macrophages, and reduced density of the endothelial fenestrae have been described in preeclamptic women [15,16,17,18]. These structural changes as well as vasoconstriction and reduced plasma volume contribute to a diminished renal filtration capacity [15,19]. The structural renal injury present during preeclamptic pregnancies seems to be reversible [19], whereas a functional impact is present postpartum and an association of PE and the risk for the development of end-stage renal disease (ESRD) later in life has been reported [20,21]. There is evidence that an undetected chronic hypertension prior to the occurrence of a preeclamptic pregnancy contributes to the increased risk for renal disease postpartum in humans [12,20,21].

The renal injury during PE pregnancies has been characterized in many animal models. Involved mechanisms in glomerular alterations include an angiogenic imbalance [9,22,23,24]. In the reduced uterine perfusion pressure (RUPP) rat model for PE, the renal impairment could be restored by vascular endothelial growth factor (VEGF) treatment [25,26]. Further, an important mechanism in preeclamptic renal injury seems to be an increased oxidative and inflammatory stress, leading to increased reactive oxygen species (ROS) production and complement activation [27,28,29,30]. Although these mechanisms cause severe renal alterations during pregnancies, there is lack of knowledge whether renal impairment is present postpartum in the PE animal model. There have been only few animal studies focusing on postpartum renal damage. In the RUPP model, mean arterial blood pressure and albuminuria were restored eight weeks after delivery. No glomerular or tubular damage was present but the glomerular filtration rate was reduced [31].

The objective of this study was to assess the postpartum effects of PE on the kidneys in a human renin-angiotensinogen transgenic rat model. In this model female rats transgenic for human angiotensinogen had normal blood pressure before and after pregnancy. They developed hypertension during the last third of pregnancy only upon mating to male rats hosting the human renin gene, as described before [32,33]. Dams developed a preeclamptic phenotype during pregnancy [34], including cardiac remodeling postpartum [35]. We hypothesized that long-term renal injury, described for PE in humans and rodent models, is mediated by high blood pressure (hypertensive crisis) during preeclamptic pregnancy and restores postpartum.

## 2. Results

### 2.1. Preeclamptic Phenotype in the Transgenic Rat Model

Pregnant transgenic female rats exhibited symptoms of PE (Figure 1A), whereas WT rats did not develop these symptoms. BP during preeclamptic pregnancy rose from day 12 to 14, following a plateau and decreased to normotensive values within two days after pregnancy. The blood pressure remained normotensive until the end of the study and four weeks postpartum (Figure 1B). WT rats were measured on single days (d0, 9, 15, 19, 28, 35, 42, 49) to enable comparison. The mean arterial blood pressure was at 105.3 ± 4.6 mmHg in WT rats during pregnancy and at 148.4 ± 20.7 mmHg in preeclamptic pregnant rats, respectively. It was significantly elevated in preeclamptic pregnancy (wild type, WT_d19_ vs. PE_d19_; *p* < 0.0001) and normalized postpartum (PE_d19_ vs. PE_d49_, *p* < 0.0001; Figure 1C). PE rats showed significantly higher urinary albumin relative to urinary creatinine during pregnancy than WT (WT_d21_ vs. PE_d21_, *p* = 0.0024), and albumin excretion was restored postpartum in PE rats (PE_d21_ vs. PE_d49_, *p* = 0.0004; Figure 1D). Although urinary albumin was higher in WT rats postpartum compared to PE rats, there was no significant difference (WT_d49_ vs. PE_d49_, *p* = 0.4917). There was an overall effect of PE on the albumin level at different timepoints (2w-ANOVA: F(1;29) = 16.15; *p* = 0.0004). 

As shown in one of our previous studies, preeclamptic rats showed reduced placental and fetal body weight compared to control rats [35], whereby maternal body weight was reduced at the end of gestation. The mean kidney weight between transgenic and WT rats was similar during (WT_d21_ = 0.8 g ± 0.06; PE_d21_ = 0.7 g ± 0.07) and after pregnancy (WT_d50_ = 0.9 g ± 0.05; PE_d50_ = 1.0 g ± 0.06) (Appendix A). Kidney weight normalized to maternal body weight showed significant higher values in the PE group compared to WT (WT_d21_ vs. PE_d21_
*p* = 0.002; WT_d50_ vs. PE_d50_
*p* = 0.001; Appendix A) This effect was mainly driven by lowered maternal body weight of the PE group, which was mainly due to the lowered number and weights of pups and placentas in the PE group [35].

### 2.2. Altered Expression of Kidney Injury Marker Genes

Genes specific for tubular injury, angiogenesis, fibrosis, tissue inflammation, or oxidative stress were selected in a hypothesis-based manner and analyzed during and after PE and WT pregnancy. Most significant alterations in renal gene expression were detected during preeclamptic pregnancy (Figure 2A). The expression of specific tubular injury markers, kidney injury molecule 1 (*Kim-1,*
Figure 2B), and *neutrophil gelatinase associated lipocalin 2* (*Ngal,*
Figure 2C) was significantly 2.5-fold increased during preeclamptic rat pregnancy. Nephrin (*Nphs-1*), which is part of the podocyte slit diaphragm, is similarly expressed in PE and WT rats during pregnancy, suggesting no damage in the slit membrane. Most genes that are associated with angiogenesis and vasoconstriction were similarly expressed during preeclamptic pregnancy in the kidney. Neither endothelin-1 (*Et-1*), vascular endothelial growth factor (*Vegf-a*), nor their receptors (*Flt-1, Flk-1, sFlt-1*) showed significantly altered gene expression during pregnancy. To identify fibrotic alterations in the kidney tissue, marker genes such as collagen type 1 and 4 (*Col-1, Col4α-1*), connective tissue growth factor (*Ctgf*) and transforming growth factor beta 1 (*Tgfβ-1*) were measured. *Col-1* and *Ctgf* were significantly altered during preeclamptic pregnancy (Figure 2A,D). The specific marker genes *CD68* (cluster of differentiation 68 molecule) and *Mcp-1* (monocyte chemoattractant protein 1) were used to characterize inflammatory processes. *CD68* expression was significantly increased during preeclamptic pregnancy compared to WT. Markers for renal oxidative stress include components of the NADPH oxidase 2 (*Ncf-1, Ncf-4, Cyb-a, Cyb-b*) and the renal-specific NADPH oxidase 4 (*Nox-4*). *Ncf-4* expression was slightly but significantly increased during PE (Figure 2E). 

When focusing on the gene expression after preeclamptic pregnancy, only few significant differences were present (Figure 2A). The inflammation marker CD68 was significantly increased after preeclamptic pregnancy compared to WT postpartum. Genes associated with nephron injury, angiogenesis and vasoconstriction, and fibrosis were similarly expressed in PE and WT rats after pregnancy. The oxidative stress marker *Ncf-4* continued to show a slightly higher expression in PE rats compared to WT postpartum (Figure 2E). In addition, the *Nox-4* expression was significantly increased in PE rats postpartum (Figure 2F). Gene expression- and *p*-values are shown in Appendix A. 

### 2.3. Mild Structural Renal Impairment

Enlargement of glomeruli and increased capillary wall thickening, the hallmark of glomerular endotheliosis, go along with albuminuria in human and rat preeclamptic pregnancy [13,17,27]. First, we assessed the glomerular damage during preeclamptic rat pregnancy and determined features of glomeruli such as size and density. The glomerular size was not changed during PE pregnancy in histologic analysis (Figure 3A). Through the characterization of the glomerular capillary area, capillary wall thickening and glomerular sclerosis could be detected. PE rats had higher percentage of periodic acid–Schiff (PAS)-positive area (WT_d21_ vs. PE_d21_, *p* = 0.0293; Figure 3B). The renal tubular system is crucial for a physiologically functioning kidney. Evaluation of the tubular damage in preeclamptic rats was performed by analyzing the tubular injury score, inflammation, and perivascular fibrosis. The tubular system was not significantly altered during PE pregnancy, yet there was a tendency of mild tubular injury during these pregnancies. Renal tubular dilation, and brush border thinning were observed (Figure 3C). Inflammation was quantified by macrophage count in the renal medulla. The comparison revealed no significant difference during pregnancy, although PE rats showed higher, yet not significant numbers of CD68-positive cells, ranging between 10 and 50 per field of view (Figure 3D). The relative fibrotic area was calculated to quantify perivascular fibrosis, though the vessels in PE and WT rats appeared similar during pregnancy (Figure 3E). 

In addition, a characterization of the glomerular and tubular damage was performed in kidneys after pregnancy of PE and WT rats. The glomerular tuft area was increased in both groups postpartum; however, in comparison to WT, glomeruli were significantly enlarged (WT_d50_ vs. PE_d50_, *p* = 0.0406; Figure 3A). The glomerular tuft area was not altered postpartum in PE rats compared to WT rats (Figure 3B). The tendency for renal tubular damage was present in kidneys from PE rats after pregnancy, although not significantly different (Figure 3C). Similar to the observations during pregnancy, the kidneys of PE rats postpartum displayed more CD68+ cells per field of view, but the comparison was not significant (Figure 3D). There were no significant differences postpartum in perivascular fibrosis between PE and WT rats (Figure 3E). 

### 2.4. Renal Function Is Slightly Altered in PE Pregnancy

The urine parameters were measured to characterize kidney filtration and excretion. Upon renal injury the reabsorption could be influenced. In a group-wise comparison, urinary creatinine was significantly reduced during and after preeclamptic pregnancy compared to WT and in the PE group on d19 compared to d49 (each comparison with *p* < 0.0001; Figure 4B). 

Similarly, sodium was significantly lower in PE d19 compared to PE d49 (*p* = 0.0263; Figure 4C). Of the electrolytes measured, only potassium and chloride concentrations were significantly affected during preeclamptic pregnancy (K^+^
*p* = 0.0063; Cl^−^
*p* = 0.0034; Figure 4D,E). In addition, potassium was significantly lower in WT d19 compared to WT d49 (K^+^
*p* = 0.0042; Figure 4D). Noticeably, PE rats had higher urine excretion compared to pregnant WT or postpartum. 

## 3. Discussion

We investigated the effect of PE on the kidney in a transgenic rat model and analyzed whether the renal alterations manifested postpartum. During PE, renal gene expression was altered. Mild glomerular alterations with increased glomerular density and tubular changes were present. Increased albumin excretion could be detected. Altered expression was present in marker genes associated with nephron injury (*Kim-1, Ngal*), fibrosis (*Col-1, Ctgf*), inflammation (*CD68*), and oxidative stress (*Ncf-4*). The expression of the tubular injury biomarker *Kim-1* [36] was elevated during PE and returned to normal after pregnancy, indicating kidney damage only during PE. Only mild 2.5-fold increase of *Ngal* expression was detected, whereby others reported a rapid 1000-fold increase upon tubular injury [37]. We observed mild changes in the expression of oxidative stress marker genes (*Ncf-4, Nox-4*) in the kidney during PE. These observations go along with studies that outline elevated ROS and impaired mitochondrial function in pregnancy in the kidney of other PE models [28,30]. It has been shown that agonistic autoantibodies (AT1-AA), which occur in the plasma of preeclamptic women, are associated with ROS production in different tissues including the kidney [28,38]. In our rat model for PE, most renal alterations recovered postpartum. Only oxidative stress marker genes (*Ncf-4, Nox-4*), mild glomerular enlargement, and decreased creatinine excretion persisted after PE. The maternal bodyweight was reduced during PE due to lower placental and fetal weights but not different postpartum as shown before [34,35]. Therefore, we assume that alterations in creatinine excretion did not result from differences in muscle mass but moreover from impaired renal function. 

We hypothesized that a preeclamptic pregnancy alone is not sufficient to predispose renal damage and influences further cardiovascular damage, as observed in humans. Most studies of other PE animal models show renal alterations during pregnancy, such as glomerular enlargement, fibrosis, glomerulosclerosis, and proteinuria [39,40,41,42]. However, only few focus on the post-pregnancy effects of PE on the kidney. In the RUPP model, BP and albuminuria recurred eight weeks after delivery, and no glomerular or tubular damage was detectable [31]. In Dahl salt-sensitive rats, glomerular and tubular injuries were present in female rats eight weeks after giving birth, detected as increased protein levels of nephrin and KIM-1 in the urine [43]. Together with our findings, these results support the evidence that persisting high blood pressure could be a cause for kidney damage observed in humans after preeclamptic pregnancy. In addition, the pressure dependency of kidney damage has been highlighted by Mori et al. They used an angiotensin II-induced model of hypertension, in which the kidney was protected from most renal injuries (fibrosis, tubular damage) when high renal arterial pressure was regulated by chronic servo-control [44]. These findings support our understanding that the pregnancy-associated renal injury in PE depends on the reversibility of pregnancy-related hypertension in PE. 

Our initial hypothesis was driven by observations in women with a history of PE, who showed a higher risk for the development of ESRD or CKD later in life [12,45]. The majority of women had no kidney function test before PE [45]. Our results show evidence that PE alone might not be sufficient to cause renal disease. Dai et al. state that women with previously existing hypertension superimposed by PE had the highest risk of developing ESRD compared to those with either pre-existing hypertension or PE [20]. Kattah et al. showed that women with a history of PE had a four times higher risk to develop ESRD and 20% of them showed decreased kidney function before pregnancy [21]. These studies underline the potential role of previously undetected hypertension resulting in superimposed PE contributing to renal disease. Moreover, we suggest that the development of renal disease following preeclamptic pregnancy is due to multiple diseases affecting the renal function. According to a multiple-hit hypothesis, two or more events would contribute to the development of renal disease and PE might be only one contributor. 

Our study displays limitations regarding the experimental procedure because we included two timepoints, at the end of pregnancy and postpartum. Organs were harvested and analyzed from both timepoints, respectively, but longitudinal BP measurement could only be performed for those sacrificed postpartum. Both groups presented comparable BP values during pregnancy. In addition, the longitudinal BP was only measured occasionally in WT to fulfil the 3R principle in animal experiments. Previous studies of our group showed that WT rats do not develop hypertension during pregnancy [34]. 

In conclusion, the transgenic rat model for PE showed mild renal impairment during pregnancy that mostly resolved postpartum. Minor renal injury with elevation of oxidative stress biomarkers and alterations of glomerular filtration were present during preeclamptic rat pregnancy. These results pointed out a slight damage of the kidney after preeclamptic pregnancy and therefore could not completely exclude PE as a risk factor for postpartum renal disease. We suspect that additional events affecting the renal function would be necessary for renal damage and subsequent failure.

## 4. Materials and Methods

### 4.1. Animal Study

Female SD rats transgenic for the human angiotensinogen gene (Tg(*hAgt*)L1623) were mated with male SD rats transgenic for the human renin gene (Tg(*hRen*)L10J) as previously described [32,35]. Female rats were controlled daily for a vaginal plug, which was presumed the first day of pregnancy (d1). Pregnant transgenic female rats resulting from the crossing of transgenic animals developed a typical preeclamptic phenotype (PE) with hypertension and albuminuria and were used for the experimental procedure [32]. SD wild-type (WT) females and males were crossed and served as a control. All rats were kept under standard conditions with controlled temperature of 22 ± 2 °C, humidity of 55 ± 15% and a 12:12 h light–dark cycle. The animals had food (Sniff V1324-300) and water ad libitum throughout the experiment. Rats were sacrificed at two different endpoints, the end of pregnancy (day 21, *n* = 9_WT_; 8_PE_) and four weeks postpartum pregnancy (day 50, n = 9_WT_; 11_PE_) by decapitation under deep isoflurane anesthesia, the latter time-point corresponding to approx. two years in humans [46]. Organs and blood were collected and stored at −80 °C until further used. The renal capsule, fat pad, and adrenal glands were detached from the kidney, which was transversely cut in half for further analysis. The study has been approved by local authorities (State Office of Health and Social Affairs, Berlin). 

### 4.2. Blood Pressure Measurement 

DSI telemetry devices (HD-T11) were implanted under inhalation anesthesia with isoflurane (isoflurane–air composite 2–2.5%) 14 days before mating as previously described [34]. Thus, the blood pressure (BP) could be measured before, during, and postpartum pregnancy in 5-minute intervals on a regular basis. Animals were single-housed during BP recordings.

### 4.3. Urine Measurement 

The 24-hour urine was collected during pregnancy on day 19 and after pregnancy on day 49 as previously described [47]. Urine samples were analyzed with the AU480 clinical chemistry analyzer (Beckman Coulter) following the manufacturer’s instructions. For albumin concentration measurement a direct, competitive ELISA (CellTrend, Luckenwalde, Germany) was performed. The amounts of chloride, potassium, and sodium were detected using ion-selective electrodes. Creatinine was enzymatically converted and colorimetrical quantified (No. OSR61204, Beckman Coulter, Krefeld, Germany). All urine parameters were normalized to either diuresis or creatinine concentration.

### 4.4. Gene Expression Analysis

Gene expression analysis was performed as previously described [48]. Briefly, frozen kidney tissue was homogenized with Qiazol Lysis Reagent (Qiagen) and Precellys ceramic beads (Peqlab Biotechnology, Erlangen, Germany). Total RNA was isolated using the RNeasy Mini Kit (Qiagen, Hilden, Germany) following the manufacture’s protocol. The High-Capacity cDNA Reverse Transcription Kit (Applied Biosystems, Waltham, USA) was used following the manufacture’s protocol to reverse-transcribe two micrograms of mRNA into complementary DNA (cDNA). To quantify the relative gene expression, a reverse-transcription quantitative polymerase chain reaction (RT-qPCR) was performed using the QuantStudio 3 Real-Time PCR System (Applied Biosystems) with either TaqMan Fast Universal PCR Master Mix or Fast SYBR Green Master Mix (both Thermo Fisher Scientific, Waltham, MA, USA). Primer and probes were designed using Primer Express 3.0 and synthesized by BioTez, Germany (primer sequences: Appendix A). Primers were diluted to a final concentration of 10 mM, probes to 5 mM. The target mRNA expression was quantitively analyzed with the standard curve method. All expression values were normalized to the housekeeping gene *18S*. The genes were selected in a hypothesis-based manner prior analysis. 

### 4.5. Immunohistochemistry

Kidney slices were fixed in 4% formalin (24 h), dehydrated, and embedded in paraffin. The kidneys were cut transversely into 2- and 4-µm-thick slices with an electronic rotary microtome (Thermo Fisher). Kidney slices were stained and scanned simultaneously using the Slide Scanner Panoramic MIDI with the objective plan-apochromat 20×/0.8× (Zeiss) to guarantee comparability as previously described [35]. The hemalaun–eosin (HE) staining was done following the manufactures protocol (T865, Carl Roth, Karlsruhe, Germany). The circumference of the glomerular-capillary tuft area within three cortical fields of view per kidney (*n* = 5) were measured to identify glomerular endotheliosis. Semi-quantitative analysis of tubular damage was performed using the previously described injury score [49]. Fifteen medullar images (200× magnification) were scored in a blinded manner regarding the percentage of tubular dilation, degeneration, necrosis, cast formation, and loss of brush border. Periodic acid–Schiff (PAS) staining was used to detect carbohydrate-rich components (glycogen) in the glomeruli. The slides were incubated in 0.5% periodic acid (Sigma Aldrich, 5 min., Steinheim, Germany ), rinsed, and Schiff’s reagent (Merck, Darmstadt, Germany ) was applied (10-min darkness). The slides were counterstained with hemalaun (Carl Roth, 5 min). The PAS-positive area of 8–10 glomerular areas per kidney (*n* = 5) was measured using the color threshold plug-in of ImageJ [50]. Masson–Goldner trichrome (MGT) staining was used to measure perivascular fibrosis. Kidney slides (2 µm) were stained in accordance with the manufacture’s protocol (3459, Carl Roth). The perimeter of the connective tissue, vessel, and lumen of three well-framed arteries per kidney were measured. The fibrotic area was calculated in relation to the total vessel area. Macrophages in the kidney tissue were stained with mouse anti-rat CD68 antibody (clone ED1; No. MCD341R, Bio-Rad, conc. 1:100) [51,52]. The secondary Cy3-conjugated donkey anti-mouse-IgG antibody (No. 715-165-151, Jackson ImmunoResearch) was used (conc. 1:300) and nuclei were stained with Vectashield mounting medium with DAPI (4′,6-diamidin-2-phenylindol; No. H-1200; Vector Laboratories, Burlingame, USA). CD68-positive cells were manually counted in ten fields of view (magnification 200×) per kidney (*n* = 5), when a nucleus was colocalized to a Cy3 signal. Histological analysis was performed in a blinded manner using the software case viewer (3D Histech Ltd., Budapest, Hungary) and Image J (National Institute of Health, Bethesda, MD, USA).

### 4.6. Statistics

Statistical analysis was performed using the software GraphPad Prism 6 and 7. Values are presented as mean ± standard deviation (SD). Results are displayed as *n* = 8–11 per treatment group and timepoint. Outliers were identified with the ROUT method using a maximum false discovery rate of 1%, values tested for Gaussian distribution and comparison of groups was done with an ordinary two-way ANOVA (analysis of variance) with a Sidak’s multiple comparison test. The f-value is indicated with respective degrees of freedom. *p*-values are indicated as * ≤ 0.05, ** ≤ 0.01, *** ≤ 0.001, **** ≤ 0.0001. 

## Figures and Tables

**Figure 1 ijms-22-03762-f001:**
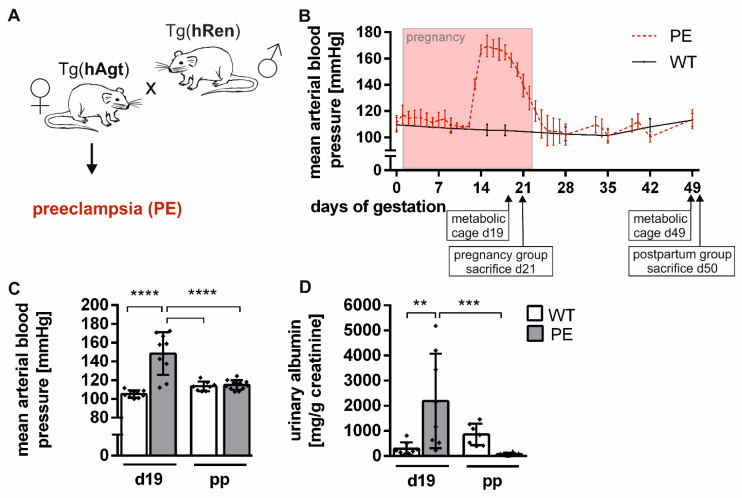
Hypertension developed during pregnancy in a preeclamptic rat model. (**A**) Female Sprague–Dawley rats transgenic for the human angiotensinogen gene (Tg(Agt)) were mated with male Sprague–Dawley rats transgenic for the human renin gene (Tg(Ren)). Thereafter, female rats developed a preeclamptic phenotype (PE). Sprague–Dawley wild-type rats (WT) were mated, and females served as a control. (**B**) Exemplary illustration of the course of the mean arterial blood pressure is shown for PE and WT over a period of 49 days, including pregnancy. It rose from day 12 to a maximum of 170 mmHg and recurred shortly after pregnancy in preeclamptic dams. Control rats did not develop hypertension. All values are presented as mean ± SD (*n* = 5–9). Timepoints of metabolic cage and sacrifice are indicated. (**C**) PE rats developed hypertension during pregnancy, mean arterial blood pressure was significantly increased (day 19) compared to WT and recovered postpartum (day 49). (**D**) Albuminuria was detected as elevated urine albumin to creatinine ratio (*n* = 8) in preeclamptic rats during (day 19) and after pregnancy (day 49) compared to pregnant WT. A statistical analysis was not conducted for the course of blood pressure. Statistical analysis was performed using a two-way ANOVA. Bars are shown as mean ± SD (*n* = 8–11). *p*-value is shown as ** *p* ≤ 0.01; *** *p* ≤ 0.001; **** *p* ≤ 0.001 (between groups).

**Figure 2 ijms-22-03762-f002:**
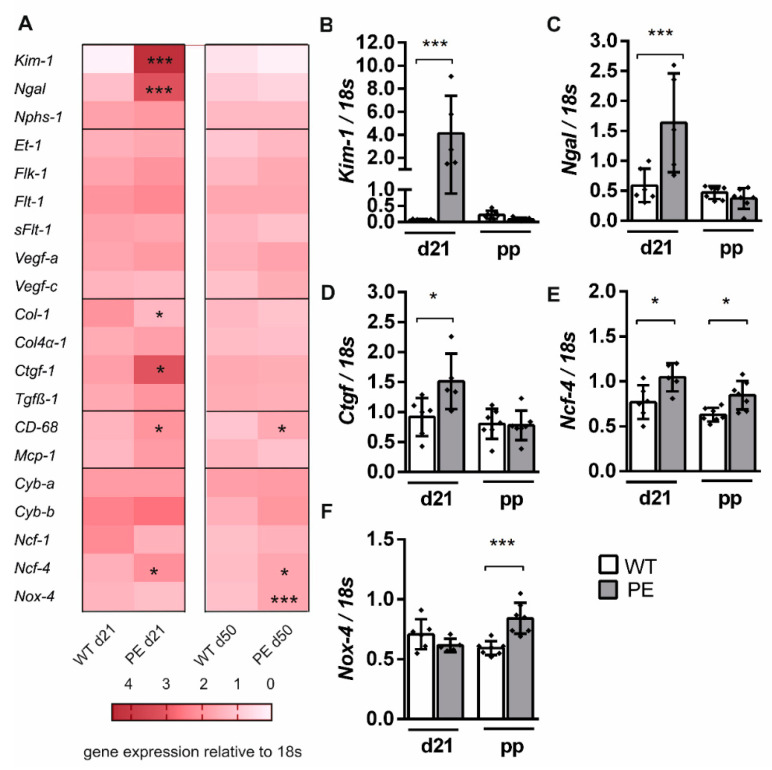
Renal injury marker expression was altered during and slightly after preeclamptic pregnancy. (**A**) Descriptive overview of hypothesis-driven relative gene expression in kidney tissue during preeclamptic (PE) and control (WT) pregnancy (day 21) and postpartum (day 50). Analyzed genes are marker for nephron injury, angiogenesis, inflammation, fibrosis, and oxidative stress (from top to bottom). Legend indicates means of relative gene expression normalized to 18S expression. (**B**–**F**) Marker genes that show altered expression in the PE group are emphasized. (**B**,**C**) Nephron injury marker kidney injury molecule-1 (*Kim-1*) and neutrophil gelatinase-associated lipocalin 2 (*Ngal*) were elevated during preeclamptic pregnancy compared to WT. (**D**) Fibrosis marker connective tissue growth factor (*Ctgf*) expression was slightly increased during preeclamptic pregnancy. (**E**,**F**) Oxidative stress marker expression of neutrophil cytosolic factor 4 (*Ncf-4*) and NADPH oxidase 4 (*Nox-4*) was higher during and/or after preeclamptic pregnancy. All genes are normalized to 18s expression. Statistical analysis was performed using a two-way ANOVA. Bars are indicated as mean ± SD (*n* = 5–7). *p*-value is shown in comparison to respective controls as * *p* ≤ 0.05; *** *p* ≤ 0.001.

**Figure 3 ijms-22-03762-f003:**
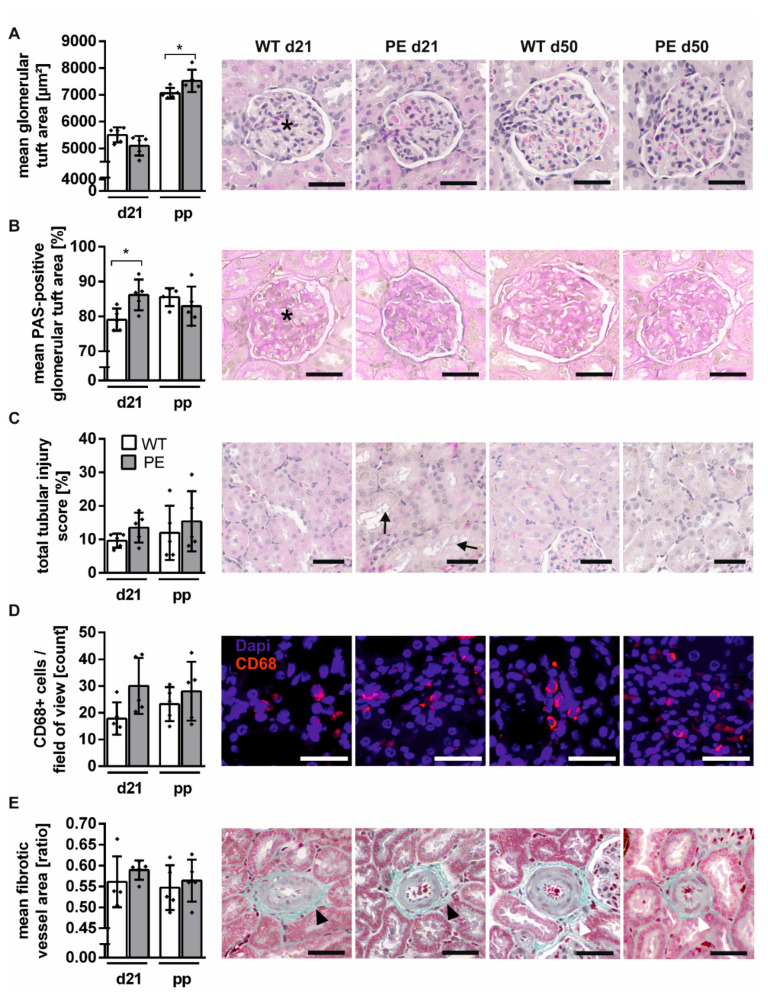
Minimal renal alterations during preeclamptic rat pregnancy. (**A**) Quantification of glomerular tuft area (asterisk) in kidneys of preeclampsia (PE) and control (WT) rats showed no glomerular enlargement during but after preeclamptic pregnancy. Representative images of hemalaun–eosin-stained glomeruli in kidney sections of PE and WT rats are shown. (**B**) The quantification of mean periodic acid–Schiff (PAS)-positive glomerular area displayed a significantly higher density in PE at day 21. Mild glomerular sclerosis was visible in the respective images of PAS-stained glomeruli. (**C**) Semi-quantitative tubular injury analysis revealed no significant difference between groups, though a slight tendency to increased damage was visible in PE at day 21. Total tubular injury score was gathered in a blinded manner in percent. Exemplary images of hemalaun–eosin-stained proximal tubules displayed mild tubular dilation and brush border thinning (arrows) at PE day 21. (**D**) Inflammation was not significantly higher in preeclamptic kidney tissue. CD68-positive cells were counted in ten fields of view per kidney. Representative images of immunohistochemically stained kidney sections for WT and PE were counterstained with DAPI (blue). Primary mouse anti-rat CD68 antibody (clone ED1) and secondary Cy3-conjugated anti-mouse antibody (red) were used. (**E**) PE did not cause perivascular fibrosis in kidney tissue. Relative fibrotic vessel area (arrow head) did not show differences between preeclamptic (PE) and control (WT) rats at all time points. Example images of Masson–Goldner trichrome-stained kidney sections displayed similar-sized connective tissue areas of the arteries. Bars are shown as mean ± SD (*n* = 5), analyzed with a two-way ANOVA. *p*-values are indicated as * *p* ≤ 0.05 (between groups). Respective images are presented with scale bars (40 µm).

**Figure 4 ijms-22-03762-f004:**
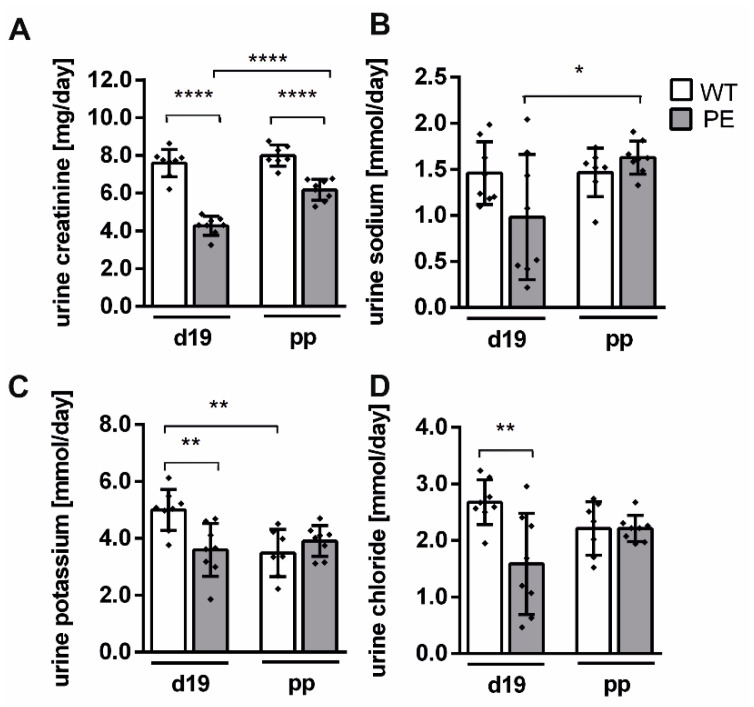
Urinary parameters were altered during PE and mostly restored after pregnancy. Amounts of urine creatinine (**A**), sodium (**B**), potassium (**C**), and chloride (**D**) are displayed per day. All parameters were reduced on day 19 of preeclamptic pregnancy (PE) compared to respective controls (WT). Urine creatinine was less reduced in PE on day 49. Sodium, potassium, and chloride recovered after PE pregnancy. Statistical analysis was done with a two-way ANOVA. Mean ± SD is indicated (*n* = 7–8). *p*-value is shown * *p* ≤ 0.05; ** *p* ≤ 0.01; **** *p* ≤ 0.0001.

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
