# Peer review of "Kidney Injury Caused by Preeclamptic Pregnancy Recovers Postpartum in a Transgenic Rat Model"

_ijms, 2021, doi:10.3390/ijms22073762_

Round 1

Reviewer 1 Report

The manuscript submitted by Kedziora and colleagues presents an examination of kidney injury in a transgenic rat model of preeclampsia. Overall the paper is quite excellent, with technically demanding and well thought-out experiments. However, there are some minor issues that should be addressed.

The ordering of data is somewhat confusing. The authors present blood pressure data in figure 1 to establish the preeclamptic phenotype, but urinary parameters are not presented until figure 4 (which is incorrectly labelled as figure 5). Since evidence of proteinuria or maternal organ dysfunction is necessary to distinguish between preeclampsia and gestational hypertension, the urinary data should be presented shortly after the blood pressure data.

Furthermore, the blood pressure data presented seems to be somewhat incomplete. Given that the authors have implanted radio-telemeters in these animals, they should be able to acquire semi-continuous readings. However, the authors only measured BP in the WT rats on a selection of 8 days. The authors later state in the discussion (lines 292-293) that BP was measured only occasionally to reduce stress on the animals, but this doesn't make much sense. Radio-telemetry allows measurement of BP with no contact between animals and handlers, so there should be no stress on the animals. Given that the PE animals had more continuous BP measurements, and experienced a large spike in blood pressure, it may be incorrectly understood that the increased measurement of BP contributed to their PE phenotype. Please explain this statement and better justify the choice to measure BP only occasionally.

Figure 1C compares blood pressure on day 19, but on line 93, the authors state that blood pressure was measured on day 18 in WT rats.

Figure 3 might benefit from having arrows or some other annotation to highlight the relevant structures.

In the discussion, the authors make two very speculative statements that are not supported by the evidence presented. First on lines 256-259 the authors suggest that postpartum renal injury in preeclamptic women is mainly due to chronic hypertension preceding pregnancy. Then on lines 276-277 the authors suspect that PE alone is not sufficient to cause renal disease. Although these statements might eventually be proven correct, there simply is not enough evidence to assert this to be true. The data presented in the paper come solely from a transgenic rat model of preeclampsia, and cannot be generalized to the human condition. These statements should be altered or removed to better reflect the limitations of the data presented here.

Reviewer 2 Report

Kedziora et al compared the blood pressure, renal gene expression, renal histology, and urine analyses between transgenetic PE rats vs WT at d19-d21 and PP in order to assess the postpartum effects of PE on the kidney. Numerous studies have shown that PE is associated with increased risk of cardiovascular diseases and end-stage kidney diseases in mothers about 10 years after PP(Turbeville and Sasser, AJP Renal physiol, 2020). It is very important to find out the renal impact by PE after PP.
The experiment was necessary and well-designed, but the authors need to elaborate how they conduct the assessments and clarify the results. 
1.    Pregnancy, itself, is a stress factor for the kidney, due to increased blood flow significantly increase the GFR. Have you thought to have a non-pregnancy age-matched group? 
2.    In Figure1, why were BP of WT and PE measured on different days? Would the BP measurement stress PE rats more than WT? At Line96, why did WTd19 compare to PE d21? But the results only show the comparison at d19 in Figure1C.
3.    At line 90-91, “BP during preeclamptic pregnancy rose from day 90 12 to 14, following a plateau and decreased to normotensive values shortly after pregnancy.” Shortly after pregnancy or after PP? 
4.    “Preeclamptic rats showed reduced placental and fetal body weight compared to control rats, whereby maternal body weight was reduced” Is this statement your results or cited from the paper? 
5.    There was no difference in kidney weight. Which kidney did you measure? Both, left, or right? Since maternal body weight was reduced in PE. Did you compare the kidney weight after normalized to the maternal body weight? Bigger animals might have bigger kidney.
6.    I am familiar delta delta Ct method to assess the gene expression result from real time RT PCR. I am aware that there are other methods to evaluate the gene expression. Please elaborate the method you used. How did you calculate the gene expression.
7.    At Line 180-181,   “The glomerular tuft area normalized postpartum in PE rats (Figure 3B)”. Since you don’t have non-pregnancy rats to compare. It is inappropriate to say “normalize”.
8.    In Figure3, you showed the results of mean glomerular tuft area, % glomerular tuft are, tubular score and CD68 cells count. How did you calculate them? Please descript how you get those results in detail in the method section. How many glumeri did you measure? 
9.    In both Figure4 and figure5, “# p ≤ 0.05 (between timepoints)”. But the line shows the comparison between WTd21 vs PEpp? I am very confused.
10.    In method section, line382, “Results are displayed as n = 5 per treatment group and timepoint.” I don’t understand. You have more than 5 animals per groups.

Round 2

Reviewer 2 Report

Thanks for replying to my comments. I have no other questions.